# Geriatric Resource Teams: Equipping Primary Care Practices to Meet the Complex Care Needs of Older Adults

**DOI:** 10.3390/geriatrics4040059

**Published:** 2019-10-21

**Authors:** Gwendolen Buhr, Carrissa Dixon, Jan Dillard, Elissa Nickolopoulos, Lynn Bowlby, Holly Canupp, Loretta Matters, Thomas Konrad, Laura Previll, Mitchell Heflin, Eleanor McConnell

**Affiliations:** 1Duke Center for the Study of Aging and Human Development, Durham, NC 27710, USA; laura.previll@duke.edu (L.P.); mitchell.heflin@duke.edu (M.H.); eleanor.mcconnell@duke.edu (E.M.); 2Duke Office of Clinical Research, Durham, NC 27710, USA; carrissa.dixon@duke.edu; 3Duke Outpatient Clinic, Durham, NC 27704, USA; janice.dillard@duke.edu (J.D.); elissa.rumer@duke.edu (E.N.); lynn.bowlby@duke.edu (L.B.); holly.causey@duke.edu (H.C.); 4Department of Case Management and Clinical Social Work, Duke University Medical Center, Durham, NC 27710, USA; 5Department of Pharmacy, Duke University Medical Center, Durham, NC 27710, USA; 6Duke University School of Nursing, Durham, NC 27710, USA; loretta.matters@duke.edu; 7Cecil G. Sheps Center for Health Services Research at University of North Carolina, Chapel Hill, NC 27516, USA; bobkonrad@gmail.com; 8Durham VA Geriatric Research, Education and Clinical Center, Durham, NC 27705, USA

**Keywords:** geriatrics, collaborative practice, geriatric workforce enhancement program, primary care

## Abstract

Primary care practices lack the time, expertise, and resources to perform traditional comprehensive geriatric assessment. In particular, they need methods to improve their capacity to identify and care for older adults with complex care needs, such as cognitive impairment. As the US population ages, discovering strategies to address these complex care needs within primary care are urgently needed. This article describes the development of an innovative, team-based model to improve the diagnosis and care of older adults with cognitive impairment in primary care practices. This model was developed through a mentoring process from a team with expertise in geriatrics and quality improvement. Refinement of the existing assessment process performed during routine care allowed patients with cognitive impairment to be identified. The practice team then used a collaborative workflow to connect patients with appropriate community resources. Utilization of these processes led to reduced referrals to the geriatrics specialty clinic, fewer patients presenting in a crisis to the social worker, and greater collaboration and self-efficacy for care of those with cognitive impairment within the practice. Although the model was initially developed to address cognitive impairment, the impact has been applied more broadly to improve the care of older adults with multimorbidity.

## 1. Introduction

The limitations of primary care practices’ capacity to care for older adults have been underscored by the Institute of Medicine, which recommended a workforce with enhanced geriatric competence and reimbursement policies that would reward effective models of care for older adults [1]. Further, the World Health Organization has advocated for the development of age friendly primary health centers accessible to older adults, employing healthcare workers well versed in geriatric syndromes and knowledgeable about community resources [2]. More recently, the Institute for Healthcare Improvement (IHI) proposed that all care for older adults be age-friendly, which they have defined as utilizing the 4Ms—What Matters, Medication, Mentation, and Mobility—to make the complex care of older adults more manageable [3].

Diagnosing dementia presents challenges for primary care practices who lack the expertise, time, and resources to perform traditional comprehensive geriatric assessment. Yet, primary care practices are caring for an increasingly older and more complex older adult patient population. In particular, 20% of those older than 65 years have mild cognitive impairment and 14% over 70 years have dementia, yet cognitive impairment is markedly underdiagnosed in primary care [4]. In its traditional form, comprehensive geriatric assessment is an interprofessional and multidimensional process that utilizes the expertise of nurses, physicians, social workers, and other health professionals to evaluate not only physical illness, but also functional status and environmental and social issues, so as to create a plan to optimize wellbeing [5]. Comprehensive geriatric assessment is the ideal process for the diagnosis and care of patients with cognitive impairment. However, it is challenging for most busy primary care practices that care for older people with multimorbidity to employ such a model.

A systematic review of the barriers primary care practices face when diagnosing and managing patients with dementia found that the barriers could be grouped into patient factors, provider factors, and system factors [6]. In particular, the studies found that primary care practices lacked essential support services, including limited access to and knowledge of community resources, lack of access to an interprofessional team to enhance management, and lack of caregiver education and support. In addition, the research found that primary care practices lacked time and sufficient reimbursement to adequately diagnose and manage patients with dementia, as well as a lack of training undermined providers’ confidence in making the diagnosis and managing subsequent care. The patient factors included stigma attached to receiving a diagnosis of dementia and delayed presentation of the patient to primary care for memory complaints. This article will describe a process by which an exemplary primary care practice developed an interprofessional and multidimensional process for the care of the older adults with mentoring from a team of geriatric experts. In particular, the practice focused on developing a method to more effectively identify and care for older adults with cognitive impairment.

## 2. Materials and Methods

The Duke Geriatric Workforce Enhancement Program (Duke-GWEP) was established to develop a healthcare workforce that maximizes patient and family engagement and improves health outcomes for older adults by integrating geriatrics with primary care [7]. To achieve this goal, we recruited primary care practices, and worked with them to create interprofessional geriatric resource teams (GRT) that could serve as a source of expertise in geriatrics and quality improvement (QI) within the practice. Each GRT was unique to the practice, but contained diverse professionals working within the practice—i.e., physicians, nurses, social worker, physician assistants, nurse practitioner, pharmacist, etc. We provided the GRTs a curriculum focused on team building, geriatric knowledge and skills, QI methods, and access to expert consultation and community resources using a hybrid learning model (Figure 1).

Previously, we found that before expecting teams to carry out QI projects, they need to establish a foundation of interprofessional collaborative practice. Therefore, GRT training began with a workshop on interprofessional collaborative practice, based on the Interprofessional Education Collaborative four core competency domains of values and ethics, roles and responsibilities, communication, and teamwork [8]. We emphasized the importance of engaging all team members in shared vision and problem solving using flexible role definitions and encouraging all team members to work at the top of their scope of practice.

Throughout the academic year, we held monthly webinars that focused on three clinical priorities specified by the GWEP funding—dementia, medication management, and care transitions—and highlighted local community resources and agencies to help address these issues (Table 1). The webinars were recorded and provided continuing education credit. The GRTs were assigned mentors from the Duke-GWEP team who guided them in choosing QI topics prior to a QI workshop that was held midway through the academic year. We encouraged practices to implement a practice improvement project focused on one of the three priority topics. The QI workshop is based on the IHI model for improvement [9]. We also provided data support and mentoring. We encouraged practices to hold monthly GRT meetings with the GRT members, the Duke-GWEP mentors, and a data support specialist. The data support specialist had expertise in public health research, health communication, project management, IT/data management, and community engagement. At the end of the year all of the GRTs gathered together to present their QI projects.

The Duke-GWEP also offered GRTs access to an Interagency Care Team (ICT): a team of geriatricians, nurse practitioners, and community partners, including clinical pharmacists and social workers who provided virtual consultations via the Electronic Health Record (EHR) for older adults with complex care needs residing in the community. This Duke-GWEP-ICT contacted the patient and family member, did a chart review, and met together to discuss the case and identify resources to help the patient remain at home. A recommendation was subsequently made to the patient and family member and to the providers at the practice.

## 3. Results

The Duke-GWEP recruited 13 practices in total over the three years, four in year 1, three in year 2, and six in year 3. To illustrate how comprehensive geriatric assessment principles are implemented in primary care, we present as a case one of the primary care practices from the first of three GRT cohorts.

### Example GRT at the Duke Outpatient Clinic (DOC)

The DOC is the major internal medicine resident teaching clinic for Duke University Medical Center, where the physician faculty and interprofessional staff recognized the challenge of caring for older adults. The clinic serves a medically and socially complex group of 4500 patients, with an average age of 62 years, many of whom are under- or uninsured. The clinic employs an interprofessional team including a licensed clinical social worker who also functions as a behavioral health specialist, a Clinical Pharmacist Practitioner, registered nurses, certified medical assistants (CMA), attending physicians, and more than 70 internal medicine residents. The DOC GRT members included the licensed clinical social worker, behavioral health specialist, Clinical Pharmacist Practitioner, attending physician and clinic medical director, and a registered nurse.

Prior to forming the GRT, there was neither a systematic approach to identify patients with cognitive impairment nor a routine process to connect patients with dementia and their caregivers to needed community resources. Barriers identified by clinic staff included a lack of expertise in the diagnosis and management of patients with cognitive impairment and a lack of time. Consequently, the evaluation of cognitive impairment often did not happen until a crisis occurred, and the default response was to refer to the geriatric specialty clinic for comprehensive geriatric assessment. Timely access to this resource was hampered by a wait time of several months to obtain an appointment, and the need to visit the large medical center across town, which resulted in missed appointments. Because many patient and family crises involved an immediate need for placement, this sometimes resulted in a hospital admission for the patient.

During the workshops, the DOC team developed both a formal vision statement and QI aim. Vision statement: “Partner with the Duke-GWEP to foster educational initiatives, interdisciplinary care teams, and collaboration with community resources to improve the care of older adult patients and their loved ones-with a specific focus on cognitive impairment.” QI project aim: “Develop and implement an interdisciplinary approach to improve care of patients and families affected by cognitive impairment.”

After developing the vision and aim statements, the clinic’s first step was workflow development, so as to screen for and diagnose cognitive impairment and clarify a process for caring for the patient once cognitive impairment was recognized. The GRT met monthly with their interprofessional team and Duke-GWEP mentors to systematically work through developing care processes to support the workflow, identify gaps, and provide additional training. The initial screening was incorporated into an ongoing project to screen for problems of social determinants of health. The team added a question to the existing screening form—“In the last two months, have you or your family had concerns about your memory or thinking?” In response to a positive answer, cognitive evaluation was then initiated using the Mini-Cog [10] performed by the CMA or the Montreal Cognitive Assessment tool (MoCA) [11] performed by the social worker, as well as for patients with concerns identified by the medical team. As the project continued, requests for cognitive evaluation began to increase from once every few months to a few times a week as providers became more aware of the clinical indicators of cognitive impairment.

The GRT developed a second workflow to guide the next steps after cognitive impairment was identified. The team obtained collateral history, performed further medical work up and treatment, documented cognitive impairment on the problem list within the EHR, and then counselled the patient and family on the diagnosis and treatment plan. An important aspect of the treatment plan was linking the patient and their caregivers to the appropriate community resources such as caregiver support programs. As cognitive impairment is now being identified earlier in the trajectory of illness, and not necessarily in conjunction with a crisis, there has been a resultant dramatic decrease in both requests for emergency placement for patients and referrals to the geriatric specialty clinic (Figure 2). The clinic is now only referring their most complex patients who can be seen more promptly since the less complex patients are managed within the primary care practice.

As part of their new process for screening, assessing, and providing care for patients with cognitive impairment, the clinic created a patient list of those with identified cognitive impairment to ensure that some of the most vulnerable patients received the services they needed. For example, they used that list to take a more proactive role in reviewing the advance care planning (ACP) needs and prioritizing these patients for ACP visits [12].

Although the model was developed to address cognitive impairment, the impact has been greater, expanding to focus on older adults with multimorbidity. The model continues to evolve and grow as other needs are uncovered. Beginning in quarter 3 of 2018, the team implemented an internal process for interdisciplinary review of complex patients, using a population health approach in which patients were identified for review by the clinical social workers based on presence of dementia and multimorbidity. The GRT modeled their processes after the Duke-GWEP’s ICT and received consultation from the Duke-GWEP nurse practitioner who developed the ICT systems. Specifically, the clinical social worker and clinical pharmacist practitioner completed a thorough review of issues regarding cognition, medication access, medication management, disease management, advance care planning, social determinants of health, labs, behavioral health/social isolation, personal/home safety, insurance/access to care, and transitions of care. The review was documented in the EHR and reviewed with the entire team including physicians and geriatrics consultants at monthly meetings.

#### Case Studies

The following case studies serve to illustrate how developing a GRT in a primary care practice enhances the capacity for conducting comprehensive geriatric assessment within primary care.

Case 1: An 88-year-old female with multiple medical problems, including recurrent GI bleeding, COPD, glaucoma, cachexia, and osteoarthritis, was abruptly left without a caregiver when her son unexpectedly died. She experienced worsening mental status and mood in the last year of her life, including an episode of delirium. Prior to establishing the GRT, this type of patient would have received a referral to the geriatrics clinic for comprehensive geriatric assessment. Instead, the primary care practice was able to manage the patient’s complex needs without a geriatrics referral. The GRT’s enhanced expertise in evaluation of her cognitive disturbance, enhanced teamwork processes, and awareness of community resources led to a timely referral to adult protective services and establishment of a new healthcare power of attorney (HCPOA) to replace her son who had recently died. Soon after, the patient experienced a serious health crisis. Her HCPOA was able to advocate effectively due to the work that had been done, and the patient was transferred to an inpatient hospice, where she died while receiving comfort care, in accordance with her wishes.

Case 2: At an acute care visit, a 63-year-old patient complained of word-finding and trouble remembering her medications. Prior to forming a GRT, the provider would have noted the concerns and advised that the patient follow-up with her primary care provider (PCP), and if the complaint persisted, a referral to the geriatrics specialty clinic would have occurred. Instead, the provider referred the patient to the GRT social worker, who was already embedded in the clinic, and able to administer the MoCA. The patient’s score was 22 out of 30; however, most of the points missed were in executive function, not memory or attention. So, the social worker, recognizing that depression can present atypically in older adults, administered a depression screen. The patient tearfully reported that her mother had died 3 years ago last week, and admitted that she has not slept well since her mother passed away, getting on average about 3 h of sleep a night. The PHQ-9 results were 19 out of 26, and after declining counseling, the patient accepted pharmacotherapy for her depression. Even though the primary diagnosis was depression, not dementia, the presence of the GRT and related protocols for cognitive impairment improved teamwork and access to staff with geriatrics expertise that, in turn, supported the diagnosis and treatment of depression which had previously been undetected.

## 4. Discussion

The barriers to the diagnosis of dementia in primary care are myriad and include provider factors, patient factors, and system factors. The GRT effectively addressed most of these barriers. Specifically, the practice was provided with training and mentorship to increase the knowledge of the providers in the identification, evaluation and management of cognitive impairment. In addition, much attention was given to linking patients and their families to community resources to support them in their homes. Further, patients were universally screened for memory concerns, addressing the barrier of delayed presentation. The time constraints and reimbursement issues remain challenging [13], but the team is better positioned to utilize all members of the interprofessional team to address some of these barriers. The GRT example and case studies illustrate effective collaborative care for patients with complex care needs, including dementia, in primary care. The members of the GRT engaged in shared problem solving; rather than a workflow that relied on one profession to identify and care for the older adult, flexible role definitions were developed that allow each profession to work at the top of their scope of practice (Figure 3).

Although not all of the team members received GRT training, the effects spread to all team members. As a result of the Duke-GWEP, the CMAs received additional geriatric training from the Duke Nurses Improving Care for Healthsystem Elders (Duke-NICHE) program [14], and for subsequent GRTs similar training was offered upfront. The Duke-GWEP training resulted in enhanced trust and confidence especially between the physicians and non-physician team members to identify and address geriatric issues. The social worker previously was brought in only during a crisis and the pharmacist was not involved. Now the pharmacist is involved in deprescribing and the social worker helps when mild cognitive impairment is identified, and an EHR-based patient list for all patients in the practice with cognitive impairment has been established so that their care needs can be anticipated and managed proactively. The CMAs are now more often included in the team discussions regarding patients and are viewed by the providers as key members of the team, resulting in empowerment and a greater sense of purpose or meaning. The practice learned from the Duke-GWEP virtual consultations performed by the ICT and was able to adopt these strategies for resource referrals to help other patients with similar problems—fall prevention, medication recommendations, and community resources.

Many programs strive to improve the care of older adults in primary care with system changes or processes with variable degrees of success [15,16]. The GRT program was unique in that there was an emphasis on team formation and webinars on geriatric principles and community resources coupled with mentoring by geriatric experts.

This example GRT from the DOC has applicability to teaching clinics in other academic health centers. As trainees rotate through these clinics that model interprofessional collaborative practice and team-based care, we expect to see improved uptake of processes that expand capacity through more effective teamwork. Team-based care that includes a variety of disciplines working to the top of their scope of practice and with expanded geriatric competency can achieve improved care of older adults without higher costs [17]. Furthermore, we believe that with the current structure of Medicare reimbursement, mentored comprehensive geriatric assessment is possible within primary care practices. Recognizing changes in cognition is a required part of the Medicare Annual Wellness Visit (G0438, G0439). As of January of 2018, Medicare provides reimbursement to providers for a comprehensive clinical visit for patients with dementia, resulting in a written care plan (CPT code 99483). This code requires an independent historian; a multidimensional assessment that includes cognition, function, and safety; evaluation of neuropsychiatric and behavioral symptoms; review and reconciliation of medications; and assessment of the needs of the patient’s caregiver. These additional billing codes provide reimbursement for practitioners for the additional time that is required in the care of these patients.

The model of shared roles allows for flexibility in the role definitions. The key is to engage all team members in shared problem solving and to make sure all of the team members are working at the top of their scope of practice. In addition, linkages with community resources are critical, as well as enhanced training in geriatrics. This example GRT illustrates sustainability since the initial training occurred over three years ago, and the practice has developed systems to sustain the program despite only one year of intensive support from the Duke-GWEP. By focusing on team formation and interprofessional collaborative practice, the teams were equipped to continue their work. We are investigating models to continue to support primary care practices in their care of older adults. This can occur through the accountable care organization or through various e-consult or telehealth programs. Without GWEP funding, teams can obtain training on geriatric care principles, QI, and interprofessional collaborative practice through a variety of professional development opportunities and organizations, such as IPEC, American Geriatric Society, and the IHI age-friendly health systems resources.

## Figures and Tables

**Figure 1 geriatrics-04-00059-f001:**
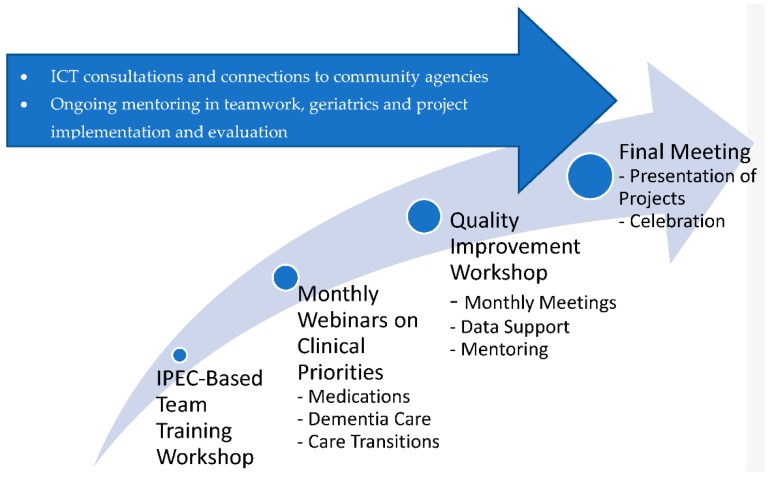
Geriatric Workforce Enhancement Program Primary Care Geriatric Resource Team Training: a year-long commitment to educational programs and process improvement activities. Abbreviations: ICT: Interagency Care Team; IPEC: Interprofessional Education Collaborative.

**Figure 2 geriatrics-04-00059-f002:**
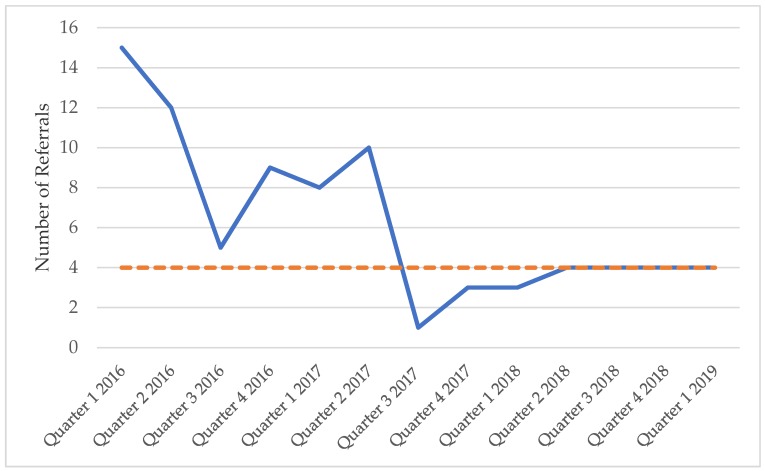
Number of referrals to the geriatric specialty clinic from the beginning of the project. The orange dotted line indicates the median.

**Figure 3 geriatrics-04-00059-f003:**
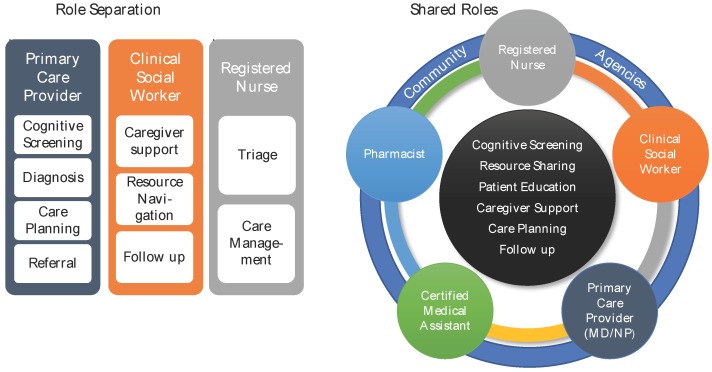
Redefining Roles and Workflow in Geriatric Primary Care before and after the establishment of a GRT.

**Table 1 geriatrics-04-00059-t001:** Geriatric resource teams (GRT) webinar topics.

Title	Objectives
Deprescribing	Explain the process of deprescribing Assess a patient’s need for deprescribing List at least two services provided by Senior PharmAssist, a community resource that provides financial assistance, medication management, community referrals, and Medicare insurance counseling
Improving Care Transitions	Define the core principles of high-quality transitions of care Describe a model for improving transitions in primary care practice Identify community resources to improve transitions of care
Improving Skilled Nursing Facility (SNF) to Home Care Transitions	Define the core principles of high-quality transitions of care from SNF to home Describe a process and model for improving SNF to home transitions that engages teams from facility, health system, community, and primary care practices Identify a role for options counselors in aiding the transitions process
Dementia: Recognition and Initial Assessment	Describe the scope and impact of dementia Develop a strategy for improving case recognition of cognitive impairment using questionnaires, structured assessments and specialty referrals Communicate effectively with older adults and families with suspected cognitive disorders Identify resources to help seniors and families cope with cognitive problems
Living with Dementia: Safety, Security, and Staying at Home	Describe the balance between autonomy and safety in caring for people with dementia List methods for improving safe management of finances and medications Develop a plan for maintaining home safety Implement measures to reduce falls among people with dementia
Medication Safety: Preventing Adverse Drug Events and Improving Transition of Care	Define and classify medication errors and preventable medication related harms Identify medication related quality measures for various care providers and settings Identify opportunities to engage community pharmacists in healthcare improvement Explain the difference between medication therapy management and medication management List at least 3 community resources for medication management
Accounting for Health Literacy in Primary Care of Older Adults	Acknowledge that healthcare is complex and problems with understanding and adherence are universal Describe the association of low health literacy with poor health outcomes Identify strategies for enhancing communication in practice to optimize a personal experience and outcomes

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
