# Peer review of "Geriatric Resource Teams: Equipping Primary Care Practices to Meet the Complex Care Needs of Older Adults"

_geriatrics, 2019, doi:10.3390/geriatrics4040059_

Round 1

Reviewer 1 Report

I enjoyed reading your manuscript. The suggestions are mainly clarification questions and suggestions for clarity.

Line 27 – one of the barriers is patient factors but do not see examples of what these are in the introduction (but they are listed in discussion section)

Line 41 – I think HRSA should be cited as I believe this statement is from their website (https://bhw.hrsa.gov/fundingopportunities/default.aspx?id=4c8ee9ff-617a-495e-ae78-917847db86a9)

Line 43 – can you please share who is on the Geriatric Resources Team? And what is the difference between the ICT and GRT? And is the DOC a GRT? Perhaps you can clarify these terms in a chart. Does Figure 3 have the GRT roles?

Line 50/chart

Suggestion to spell out ICT and IPEC since this is the first time thy are seen.

Monthly webinars – suggestion to change title to “Clinical Priorities” to be consistent with text and not have the topics numbered and rather be bulleted like the others since there were more than 3 webinars done. And are these clinical priorities established by Duke overall, DOC or the GRT?

Quality Improvement Workshop – suggestion to change to plural: workshops since multiple

Also noticed there isn’t mention of the final presentation of projects or celebration which you may want to discuss as part of the discussion. Were there any PDSA cycles or changes to the GRT over time?

Line 71/table

What is Senior PharmAssist?

Suggestion to spell out acronyms, like PC in Improving SNF to Home Care Transitions

Line 82 – would be interesting to know how many cohorts there were of the GRT.

Line 101 – are the face to face workshops the IPEC or Quality Improvement workshops or something else?

Line 104 – suggestion to use “older adult” instead of elderly (although I know this is part of the vision statement)

Line 113 – was everyone given the Mini-cog or only if they said yes to concerns about memory? Suggestion to add something like “based on positive screening result, mini-cog administered”

Lines 128-129 – noticed that “more” is stated 3 times in this sentence

Figure 2 – did something happen between quarter 2 2018 and quarter 1 2019 or was it consistently at 4 referrals?

Line 141 – suggestion to give more specific time frame than “over past 6 months”

Line 220 – which case example does this refer to? Or both?

Author Response

Response for Reviewer 1 comments

Line 27 – one of the barriers is patient factors but do not see examples of what these are in the introduction (but they are listed in discussion section)

Response: We have added the patient barriers to the introduction.

Line 41 – I think HRSA should be cited as I believe this statement is from their website (https://bhw.hrsa.gov/fundingopportunities/default.aspx?id=4c8ee9ff-617a-495e-ae78-917847db86a9)

Response: We have added the reference.

Line 43 – can you please share who is on the Geriatric Resources Team? And what is the difference between the ICT and GRT? And is the DOC a GRT? Perhaps you can clarify these terms in a chart. Does Figure 3 have the GRT roles?

Response: We have clarified the composition of the GRTs by inserting the sentence – ‘Each GRT was unique to the practice, but would contain diverse professionals working within the practice – i.e. physicians, nurses, social worker, physician assistants, nurse practitioner, pharmacist, etc.’ We have also clarified that the example GRT is from the DOC by expanding the title of the section to read ‘3.1. Example GRT at the Duke Outpatient Clinic (DOC)’. I have specified the membership of the DOC GRT. We have made sure that the ICT is clearly explained in the paragraph beginning on line 83.

Line 50/chart:

Suggestion to spell out ICT and IPEC since this is the first time they are seen.

Response: We added a key to the abbreviation to the title since adding the long names would have been difficult within the figure.

Monthly webinars – suggestion to change title to “Clinical Priorities” to be consistent with text and not have the topics numbered and rather be bulleted like the others since there were more than 3 webinars done. And are these clinical priorities established by Duke overall, DOC or the GRT?

Response: We have made the suggested changes to the figure. The clinical priorities were established by the GWEP funding and we have added this clarification to the text.

Quality Improvement Workshop – suggestion to change to plural: workshops since multiple

Response: There was only one Quality Improvement workshop, therefore we have left it singular in the figure.

Also noticed there isn’t mention of the final presentation of projects or celebration which you may want to discuss as part of the discussion. Were there any PDSA cycles or changes to the GRT over time?

Response: We have added the final presentation to the methods section. The paper focuses on the experience of one GRT at the DOC. The GRTs completed a QI project and PDSA cycles, but there were no changes to the GRT program over the course of the year.

Line 71/table:

What is Senior PharmAssist?

Response: A description of Senior PharmAssist has been added to the table.

Suggestion to spell out acronyms, like PC in Improving SNF to Home Care Transitions

Response: The acronyms have been spelled out.

Line 82 – would be interesting to know how many cohorts there were of the GRT.

Response: There were 3 cohorts. This information has been added.

Line 101 – are the face to face workshops the IPEC or Quality Improvement workshops or something else?

Response: The face-to-face workshops were the same as the IPEC and QI workshops. The words face-to-face were removed because they did not add to the understanding.

Line 104 – suggestion to use “older adult” instead of elderly (although I know this is part of the vision statement)

Response: This has been done.

Line 113 – was everyone given the Mini-cog or only if they said yes to concerns about memory? Suggestion to add something like “based on positive screening result, mini-cog administered”

Response: The suggested language has been incorporated.

Lines 128-129 – noticed that “more” is stated 3 times in this sentence

Response: The sentence has been revised so that more only appears once.

Figure 2 – did something happen between quarter 2 2018 and quarter 1 2019 or was it consistently at 4 referrals?

Response: The number of referrals between quarters 2 2018 and quarter 1 of 2019 was consistently at 4.

Line 141 – suggestion to give more specific time frame than “over past 6 months”

Response: This has been done. The time frame was stated in terms of the quarter that the ICT was started.

Line 220 – which case example does this refer to? Or both?

Response: This statement was referring to the DOC GRT as a whole. The sentence was clarified.

Reviewer 2 Report

I have many remarks on this paper:

-The mean age of the selected patients is 62 years: now geriatric patients have a mean age of 86 years.

-The knowledge of geriatrics is in general hospitals not better than in the primary care.

-detecting of geriatric syndromes is the first task og the GP and the primary care; fine-tuning and therapeutic planning is the task is the task of the geriatric unit in the general hospital;

-I see the term "frailty" never appearing: now the real hot topic, so I think this paper is really not acceptable;

-In the DOC's I see there are only internists: the internist have no knowledge of geriatric medicine!!

-In case 1: a good example of wroing geriatric practice. Sending a geriatric patient to a nursing facility without first geriatric evaluation is now ethical not longer acceptable;

-case 2: age of the patient is 63 years...: this was a patient for the internist or Psychiatry.

Author Response

The reviewers points are well taken. However this paper was a case study in a primary care practice. Hospitals were not addressed. Similarly, though frailty is an important subject, our work focused on dementia. It is true that Internists do not have in depth geriatric knowledge; nevertheless, our goal was to establish an organizational framework in which they could competently diagnose and treat patients with dementia. We feel both case examples were well suited to illustrate improved geriatric knowledge. 

Reviewer 3 Report

This is a very nicely written and professionally presented paper that would make a very valuable contribution to the special issue theme. I have no substantial criticisms of the paper. In table 1 (on p. 3) it may be useful to spell out in full SNF where it first occurs (for the benefit of readers from outside the US who may not be familiar with this abbreviation). I strongly recommend publication.

Author Response

Response to Reviewer 3

In table 1 (on p. 3) it may be useful to spell out in full SNF where it first occurs

Response: This has been done